# Effects of Climate Change on the Habitat of the Leopard (*Panthera pardus*) in the Liupanshan National Nature Reserve of China

**DOI:** 10.3390/ani12141866

**Published:** 2022-07-21

**Authors:** Jinyuan Zeng, Jie Hu, Yurou Shi, Yueqi Li, Zhihong Guo, Shuanggui Wang, Sen Song

**Affiliations:** 1School of Life Sciences, Lanzhou University, No. 222 of South Tianshui Road, Lanzhou 730000, China; zengjy20@lzu.edu.cn (J.Z.); huj19@lzu.edu.cn (J.H.); shiyr21@lzu.edu.cn (Y.S.); liyueqi3068@163.com (Y.L.); 2Liupanshan National Nature Reserve Administration, Guyuan 756401, China; gzh13895440002@163.com (Z.G.); wsg13909548327@163.com (S.W.)

**Keywords:** MAXENT model, future climate, different geographic scales, suitable habitat distribution, population protection

## Abstract

**Simple Summary:**

Climate change affects animal populations by affecting their habitats. The leopard population has significantly decreased due to climate change and human disturbance. We studied the impact of climate change on leopard habitats using infrared camera technology in the Liupanshan National Nature Reserve of Jingyuan County, Ningxia Hui Autonomous Region, China, from July 2017 to October 2019. We captured 25 leopard distribution points and used the MAXENT model to predict and analyze the habitat. We studied the leopard’s suitable habitat area and distribution area under different geographical scales in the reserve. Changes in habitat area of leopards under three climate models in Guyuan in 2050 were also studied. We conclude that the current main factors affecting suitable leopard habitat area were vegetation cover and human disturbance. The most critical factor affecting future suitable habitat area is rainfall. Under the three climate models, the habitat area of the leopard decreased gradually because of an increase in carbon dioxide concentration.

**Abstract:**

Climate change affects animal populations by affecting their habitats. The leopard population has significantly decreased due to climate change and human disturbance. We studied the impact of climate change on leopard habitats using infrared camera technology in the Liupanshan National Nature Reserve of Jingyuan County, Ningxia Hui Autonomous Region, China, from July 2017 to October 2019. We captured 25 leopard distribution points over 47,460 camera working days. We used the MAXENT model to predict and analyze the habitat. We studied the leopard’s suitable habitat area and distribution area under different geographical scales in the reserve. Changes in habitat area of leopards under the rcp2.6, rcp4.5, and rcp8.5 climate models in Guyuan in 2050 were also studied. We conclude that the current main factors affecting suitable leopard habitat area were vegetation cover and human disturbance. The most critical factor affecting future suitable habitat area is rainfall. Under the three climate models, the habitat area of the leopard decreased gradually because of an increase in carbon dioxide concentration. Through the prediction of the leopard’s distribution area in the Liupanshan Nature Reserve, we evaluated the scientific nature of the reserve, which is helpful for the restoration and protection of the wild leopard population.

## 1. Introduction

Habitat is a critical requirement for animal survival and reproduction, providing food, protection, and a living environment [1]. Shrinking of biological habitat areas can lead to the loss of biodiversity [2]. Industrial development and global climate change are gradually affecting species habitats. For species that do not adapt to climate change, extinction may be the result [3]. The habitats of many organisms are threatened, and some wildlife with small populations are at risk of extinction [4]. Many researchers have built stable and accurate species habitat prediction models [5]. These models can predict species distributions, identify and promote needed protected areas, help restore the populations and genetic diversity of endangered species, and reduce the rate of species extinction [6,7,8].

Species distribution models have been used in many research fields. For species distribution prediction, the commonly used species distribution prediction models include CLIMEX, GARP, BIOCLIM, DOMAIN, GLM, ANN, RF, and MAXENT [9]. The MAXENT model has produced the best results among these models, particularly when known records of a target species are sparse. The prediction principle calculates the distribution law of species under different environmental conditions according to the known species distribution information and environmental variables. The model has advantages such as better simulation results when the data set is incomplete and reduced demand for a large sample size [10,11,12]. Better prediction results can be obtained if the sample size is greater than 10 [13]. The model has simple operation, with high prediction accuracy and good stability [14]. Many researchers have tested the operating conditions of the model, and this is helpful for future researchers to use the model for predictions that are consistent with realistic conditions [15]. MAXENT is widely used to study the impact of climate at different geographical scales and times on species distribution, species invasion monitoring, natural disasters, habitat suitability evaluation, conservation biology, and biogeography [8,16,17].

The leopard (Panthera pardus) is a large and rare species of carnivorous cat that is widely distributed in Africa, the Middle East, Central Asia, the Indian subcontinent, Southeast Asia, Russia, and China [18,19,20]. The North China leopard (*Panthera pardus japonensis*) is a subspecies widely distributed in eastern, central, and southern China, mainly in mountain areas, thickets, and forests with good concealment [21]. Due to overhunting, habitat destruction, and populations being small and isolated from each other, the total number of leopards has sharply declined. This is a national first-class protected animal and listed as endangered by the IUCN list [22], but there are relatively few studies on leopards in China. Only in Shanxi, Henan, Shaanxi, Ningxia, and other provinces have conservation personnel conducted adequate studies on the number and distribution of the wild leopard population [23,24]. Few individual leopards have been reported in other areas, and the current number of leopards in China is estimated to range from 174 to 348 [25]. Some studies have focused on the genetic diversity and intestinal microorganisms of leopards [26,27,28], but there are fewer studies on the habitat distribution and behavior of leopards in the field.

A flagship species in the ecosystem can be used as a signal to protect the ecosystem [29]. When the underlying ecosystem has recovered, the number of the flagship species usually increase [30]. The leopard is the flagship species of Liupanshan National Nature Reserve, and their numbers reflect the current status of the ecosystem. Leopard protection can help control the populations of other organisms in the area, maintain the regional ecosystem balance, and prevent the overloading of environmental resources.

By understanding the habitat distribution and influencing factors of the leopard, we can provide a basis for its habitat protection. This information can help determine critical protection areas and the formulation of conservation strategies. In this study, we used infrared camera technology to study the distribution of leopards in Liupanshan National Nature Reserve. Leopard habitat was predicted under different environmental factors. The goal was to provide a reference for the work of the reserve by predicting the suitable distribution areas and providing technical support for the protection of leopards in the Liupanshan population.

## 2. Research methods

### 2.1. Study Area

The study area was the autonomous region of the city of Guyuan, China, with a total area of 900.71 km^2^. The average altitude of the area is 2300–2600 m, and the highest altitude is 2942 m [31]. Important tributaries of Weihe River, Jinghe River, Qingshui River, and Hulu River originate here. They play an essential role in conserving water sources and improving the regional environment and are ecological barriers that protect against the semi-arid areas in Northwest China. Liupan Mountain is in a transitional zone between semi-humid and semi-arid regions, with the climatic characteristics of continental and marine monsoon margin. The average annual temperature is 5–6 °C, and the average annual rainfall is 600–700 mm [31]. The climate of the protected area differs from that of Northwest China, with precipitation and humidity higher than the average level of Northwest China [32]. The climate in this area is suitable for the survival of many wild animal species. Common animals in the region are wild boar (*Sus scrofa*), squirrel (*Sciurotamias davidianus*), hog badger (*Arctonyx collaris*), Tolai hare (*Lepus tolai*), muntjac (*Elaphodus cephalophus*), roe (*Capreolus pygargus*), and mainland Reeve’s muntjac (*Muntiacus reevesi*) [33]. Many small and medium-sized wild animals in this area provide sufficient food for large carnivores, and this is conducive to their survival and reproduction.

### 2.2. Species Distribution Data

The data were obtained from a survey of animals in the Liupanshan National Nature Reserve of Ningxia through the infrared camera trap method from 2017 to 2019. Using the kilometer grid method requirement, we installed 60 infrared cameras [34] in the Liupanshan Nature Reserve. The infrared camera method recorded the longitude and latitude of the leopard distribution points and obtained 25 distribution points. To avoid the occurrence of spatial autocorrelation, the distribution points were checked and screened using the buffer analysis method [35], and the spatial resolution of environmental data was 30”. Therefore, the radius of the buffer zone was set to 0.5 km, and only one of the distribution points was retained when the distance was less than 0.5 km. A total of 19 effective species distribution sites were obtained (Figure 1). The distribution points were transferred to Excel, converted, and stored in CSV format, which was applied to the model calculations.

### 2.3. Environmental Data

The environmental data used in this study included bioclimatic, human interference, terrain, vegetation, and soil factors (Table 1). Climate factors were obtained from the world climate data website WordClim. The data principle was to generate global raster data from 1970 to 2000 and extract 19 bioclimatic factors [36] from the study area by GIS. Human interference factors, obtained from the National Primary Geographic Center, provided the 2009 1:1 million geographical data, road density, residential density, and water sources. The distances between species distribution and roads, settlements, and water sources were calculated using the Euclidean distance. The resolution from the NOAA website was used to extract altitude, aspect, and slope data from the 30 m × 30 m elevation map as terrain factors, and the spatial analysis was conducted in ArcGis10.4. Data on vegetation and soil factors were obtained from the information ring of the Chinese Academy of Sciences. The 2018 data mainly included vegetation cover class degree FVC, normalized vegetation index NDVI, and land-use type. In GIS 10.4, through mask extraction and resampling operations, the environmental data of the study area were obtained for subsequent model construction.

### 2.4. Screening of Environmental Variables

Many environmental factors affect the distribution of leopards. According to the living habits of leopards, a total of 28 environmental factors such as bioclimatic factors and human interference factors were used to construct a preliminary model for the prediction of suitable areas for leopards at different geographical scales. To avoid over-fitting of the prediction model, we eliminated the environmental factors with a high correlation among the 28 environmental factors [14]. IBM SPSS Statistics 26 software was used to analyze the correlations among the remaining environmental data, and the environmental factors with a correlation greater than 0.7 were screened and removed. Under the three geographical scales of Liupanshan National Nature Reserve, Guyuan, and Ningxia Hui Autonomous Region, nine climatic factors were retained [37] (Table 2). We used the jackknife method to test the contribution of environmental variables to the model [38]. The main influencing factors that contributed most to the model were screened out and used for the current Liupanshan National Nature Reserve at the scale of Guyuan and the scale of Ningxia Hui Autonomous Region to construct a leopard habitat prediction model.

The global climate model (GCM) IPPC5 climate prediction from the WordClim website was used to explore the impact of future climate on leopard survival. To build a preliminary model, we selected three approaches—rcp2.6, rcp4.5, rcp8.5—and their 19 bioclimatic environmental factors [39]. For the environmental factors with too large a contribution value or almost no contribution value, we used IBM SPSS Statistics 26 software to analyze the correlation of the remaining environmental data and select and remove the environmental factors where the correlation was more than 0.8 to ensure that the number of environmental factors was not too small, resulting in a lack of features and reduced accuracy [40]. Four climate factors were retained under the contemporary climate model, and five climate factors were retained in the future climate models, which were used to build the habitat prediction model of the leopard. The contributions of environmental variables to the model were tested by the jackknife method, and the main influencing factors that made a significant contribution to the model were selected.

### 2.5. Results Evaluation

We loaded the environmental data and leopard distribution data into the MAXENT model. We then used 75% of the leopard coordinate data for training operation and 25% of the data for verification, repeating this 10 times [41]. The simulation test results were tested by the ROC curve and the evaluation criteria: the area surrounded by the ROC curve and abscissa, that is, the AUC value, is 0–1, and the AUC value is closer to 1, indicating that the more significant the correlation between the environmental variables and the distribution model, the better the prediction result [42,43,44]. The MAXENT model can generate the ROC curve and the corresponding AUC value, extract the corresponding value, and redraw the image in IBM SPSS Statistics 26.

### 2.6. Classification of Suitable Birth Grades

The suitable zone value of the leopard output generated by MAXENT was between 0 and 1, and the closer the value was to 1, the more likely leopards existed in the zone. The MAXENT results were reclassified in GIS, and the habitat was manually divided into two grades: suitable habitat and unsuitable habitat. The habitat threshold was decided by a logstic10 operation threshold file automatically generated by the MAXENT model [45].

## 3. Results

### 3.1. Prediction Results of the Model

Three scales of suitable habitats were predicted for Liupanshan National Nature Reserve, Guyuan, and the Ningxia Autonomous Region (Figure 2A). The results of the ROC curve of Liupanshan showed that the average AUC of 10 training sessions was 0.841, indicating that the prediction result of the MAXENT model was good, and the model could better predict the habitat of leopard in Ningxia Liupanshan National Nature Reserve. The AUC values of the scale models in Ningxia and Guyuan were higher (Figure 2B,C), reaching 0.992 and 0.986, respectively, indicating that the prediction result of the MAXENT model is accurate and that the model can reasonably predict the suitable habitat of the leopard in Ningxia and Guyuan.

According to the prediction results of different geographical scales, the loss of local suitable region prediction caused by too small a prediction area, a large prediction area, and a meaningless, high AUC value prediction area were obtained, so we chose to predict the future habitat of the leopard at the geographical scale of Guyuan [46]. The model accuracy was verified for different climate model predictions in the modern and rcp2.6, rcp4.5, and rcp8.5 model suitable areas (Figure 3). The ROC curve was trained 10 times, and the average AUC values of the training set were 0.985, 0.982, 0.987, and 0.988, respectively, indicating that the prediction results of the MAXENT model were good. The model could reasonably predict the suitable habitat distribution of the leopard in the future climate model.

### 3.2. Distribution and Area of the Potentially Suitable Growth Area of the Leopard

The threshold setting of the suitable growth zone of the MAXENT prediction model was obtained from the threshold mean file obtained by running the model 10 times, and the corresponding areas were divided into suitable growth areas and non-suitable growth zones. The threshold settings of the three areas were 0.4991, 0.4407, and 0.3748. According to the prediction results of the model (Figure 4), under the three predicted scales of Liupanshan National Nature Reserve, Guyuan, and Ningxia Hui Autonomous Region, the suitable habitat distribution area of the leopard was 77.0914 km^2^, 106.7117 km^2^, and 211.3423 km^2^, respectively. The proportion of land area to the predicted area was 18.28%, 10.16%, and 0.32%, respectively.

The habitat area of Guyuan was divided into suitable habitat and unsuitable habitat (the threshold was 0.337) based on the logstic10 threshold file generated by the MAXENT model, which was used to analyze and explore the effects of different environmental and climatic factors on the habitat area change of the Chinese leopard in 2050. In the modern climate (Figure 5A), the suitable habitat area of the leopard is 286.9318 km^2^. Under the rcp2.6 climate model (Figure 5B), the core area of the habitat of the leopard is 152.3398 km^2^, accounting for 1.45%; the entire non-suitable area, 101.3167 km^2^, accounts for 96.49%; the area with 79.3883 km^2^ expands its habitat, accounting for 0.76%; and the area with 136.6051 km^2^ shrinks, accounting for 1.30%. Under the rcp4.5 climate model (Figure 5C), the core area of the habitat of the leopard is 151.6245 km^2^, accounting for 1.44%; the entire non-suitable area, 10,149.55 km^2^, accounts for 96.66%; the area of 61.5081 km^2^ has expanded its habitat, accounting for 0.59%; and the area of 137.3203 km^2^ has shrunk its habitat, accounting for 1.31%. Under the rcp8.5 climate model (Figure 5D), the core area of the leopard habitat is 117.2945 km^2^, accounting for 1.11% of the unsuitable area of 10,176.01 km^2^, accounting for 96.91%; the area with 35.0453 km^2^ has expanded its habitat, accounting for 0.33%; and the area with 171.6504 km^2^ has shrunk its habitat, accounting for 1.63%.

### 3.3. Contribution Analysis of Environmental Variables to the Model

The contribution analysis of environmental variables in Liupanshan National Nature Reserve on the geographical scale (Table 3) shows that the total contribution rate of the four environmental factors bio6, bio3, aspect, and vfc reached 86.00%. They provide the greatest contribution rate to the leopard habitat model, which is the main factor affecting the leopard’s habitat; less than 20% is contributed by the other five environmental impact factors. Contribution analysis of environmental variables in Guyuan showed that the total contribution rate of the four environmental factors of ndvi, vfc, bio11, and bio12 reached 86.50%, and the contribution rate of the model was the largest, which was the main factor affecting the habitat area of the leopard. The contribution analysis of environmental variables in the Ningxia Autonomous Region showed that the total contribution rate of the three environmental factors was based on analysis of the different climate models rcp2.6 and rcp8.5 (Table 3). The contribution analysis of environmental variables under the rcp2.6 model showed that the total contribution rate of the environmental factors bio16 and bio14 reached 84.2%, which is a considerable contribution and is the main factor affecting the leopard habitat. Contribution analysis of environmental variables under the rcp4.5 model showed that the total contribution rate of the bio16 and bio17 environmental factors reached 79.2%, making them the most significant contributors and the main factors affecting the leopard habitat. The contribution analysis of environmental variables under the rcp8.5 model showed that the total contribution rate of bio14 and bio16 environmental factors reached 67.9%, and the contribution rate of the model was the largest, indicating that these are the main factors affecting the leopard habitat.

### 3.4. Jackknife Test Analysis of Environmental Factors

On a geographical scale, the jackknife test of the habitat model of the nature reserve showed that the environmental factors bio6 and bio12 had the highest gain when used alone (Figure 6A), and the bio6 environmental factor had the most significant impact on the leopard habitat. There was a difference between the jackknife test and the variable contribution analysis, which showed an interaction between multi-environmental and single environmental factors. The jackknife test of the Guyuan habitat model showed that the gain of environmental factor bio12 (Figure 6B), vfc, was the highest when used alone, and that the bio12 vfc environmental factor had the greatest impact on the leopard habitat. The jackknife test of the habitat model in the Ningxia Autonomous Region showed the gain of the environmental factor bio12; bio2 was highest when used alone (Figure 6C), and the bio12 environmental factor variable had the most significant impact on the leopard habitat.

The analysis was carried out under the different climate modes of rcp2.6, rcp4.5, and rcp8.5. The jackknife test of the habitat model under the rcp2.6 model showed that the environmental factors bio16 and bio15 alone had the highest gain, and the bio16 environmental factor was the variable that had the greatest impact on leopard habitat (Figure 7B). The jackknife test of the habitat model in the rcp4.5 model showed that the gain of the environmental factors bio16 (Figure 7C) and bio17 were the highest when used alone, and the environmental factors of bio16 and bio17 had the greatest impact on the leopard habitat. The jackknife test of the habitat model in the rcp8.5 model showed that the gain from environmental factors bio16 and bio15 was the highest when they were used alone (Figure 7D), and environmental factor bio16 had the greatest impact on the leopard habitat.

## 4. Discussion

### 4.1. The Impact of Different Geographical Scales on the Leopard’s Habitat Distribution

The leopard is mainly distributed in the broad-leaved forest and coniferous and broad-leaved mixed forest habitat of the Liupanshan National Nature Reserve. There are also records of leopards in the Liupanshan Provincial Nature Reserve of Jingyuan County. Other literature and media reports show no records of leopards in other areas of Ningxia [47]. However, the results of this study predicted, at a geographic scale (Figure 4), that a suitable habitat for leopards in the Ningxia Autonomous Region is concentrated near the Liupanshan National Nature Reserve in Jingyuan and Longde County, and there is a low probability of leopards existing in the Liupanshan provincial reserve. The probability of the existence of leopards is confirmed by previous records [47]. The establishment of the Liupanshan National Nature Reserve protects leopard habitat and provides a place for its survival and reproduction. With expansion of the geographical scale, the area suitable for the leopard obtained by the model increased, indicating that there are potentially suitable habitats for leopards in the Ningxia Autonomous Region. This needs to be verified by actual monitoring.

The geographical-scale habitat prediction of Ningxia Autonomous Region shows that the distance from roads and the distance from human settlements and disturbance are important factors that limit the habitat expansion of the leopard. When human interference factors are not excluded in the model’s prediction, the distance between the distribution site of the leopard and the distance from the road and residential area account for more than 90% of the contribution rate to the model. After removing the human interference factors, vegetation coverage becomes the main influencing factor (Table 3). Geographic-scale habitat prediction in Guyuan shows that ndvi, vfc, and bio12 are the main factors affecting leopard habitat (Table 3) and indicate that leopard habitat requires specific vegetation coverage in vigorous forests. In addition, without considering vegetation coverage, the predicted area of suitable habitat for leopards increased by nearly 1.8 times (Figure 5), which also shows that a particular vegetation coverage has a more significant impact on the distribution of suitable habitats for leopards. Forests provide abundant food and habitat for herbivores, which promotes the growth of their populations and in turn provides sufficient food for leopards [48,49,50]. Lush forests also provide good cover for leopards, helping to improve their hunting success rate. Trees provide an excellent natural barrier for leopards to raise their cubs during breeding, and tree cover significantly reduces the possibility of being discovered by other predators. Another critical factor is that the annual rainfall also affects the growth of forest vegetation and thus affects the distribution of suitable leopard habitats [51]. The scale prediction of Liupanshan National Nature Reserve shows that bio6 is an essential factor affecting its habitat distribution, except for vegetation cover. This may be related to the leopard’s low food consumption, high energy consumption, and higher survival cost in winter.

### 4.2. The Impact of Future Climate on the Leopard’s Habitat Distribution

Under the future climate model, the prediction results of the suitable habitat for leopards based on the MAXENT model show that bio16 is the most important environmental factor affecting leopard habitat under the three climate models in 2050 (Table 3), followed by bio14, bio15, and bio17. This indicates that rainfall has a significant influence on the suitable habitat distribution of leopards. Rainfall affects the growth of vegetation and indirectly affects leopard food sources and shelter. The decrease in rainfall in dry and humid seasons leads to the contraction and loss of suitable leopard habitat [52]. In terms of habitat change, the core distribution area of the leopard in the Ningxia Autonomous Region under different climate models is concentrated in the Liupanshan National Nature Reserve. The suitable habitat area is expanded to the Liupanshan Provincial Nature Reserve (Figure 5). Under the rcp2.6 model, the currently suitable habitat area is predicted to be 286.9318 km^2^, and the suitable habitat area will be reduced to 57.2168 km^2^ in 2050, accounting for 19.94%. In rcp4.5 mode, the suitable habitat area will be reduced 75.8122 km^2^, accounting for 26.42%. Under the rcp8.5 model, the suitable habitat area will be reduced 136.6051 km^2^, accounting for 47.61%. These results show that as global carbon dioxide emissions rise, the total area of habitat suitable for leopards will shrink, and most losses will occur in Liupanshan National Nature Reserve. The Liupanshan Provincial Reserve will become a suitable habitat expansion area for leopards (Figure 5). The loss of suitable habitat appears to be related to the increase in carbon dioxide content affecting changes of temperature and rainfall in the suitable area. These changes are not conducive to the survival of existing vegetation and affect changes in vegetation community structure. This results in a decrease in the number of herbivores in this area and indirectly affects the food sources of leopards [53,54]. In addition, a decrease in vegetation areas in the leopard’s habitat will lead to a decrease in leopard camouflage and decreased hunting success. A sparse vegetation environment is also not conducive to leopards feeding their cubs, and breeding costs will increase [55].

### 4.3. Protection Suggestions for Leopard Habitat in the Reserve

The potentially suitable area for leopards is beyond the scope of the nature reserve. We suggest that the reserve consider expanding the protected area, promoting the restoration of vegetation, and providing an improved habitat for leopards. Ecological corridors could also be constructed to increase the connectivity between national and provincial nature reserves and provide a route for the leopards to enter suitable habitats.

Human disturbance limits the expansion of the suitable area for leopards in Ningxia. The distance from the road and distance from the settlement in the model are the most critical factors affecting the suitable habitat of leopards on the geographical scale in Ningxia. In practice, human interference can involve tourism, digging for medicinal plants, and collection of wild vegetables. These activities can have a significant impact on the daytime activities of leopards and other wild animals distributed in Liupan Mountain, and this needs to be publicized to reduce the activities of humans in areas with leopard activities. Tourists are advised to travel using prescribed routes and not venture deep into the interior of the reserve. The road formed by the infrastructure restricts the activity area of leopards and the road sounds can disturb leopards. For the core area of the leopard habitat, road construction should be minimized, and mountains should be afforested. In addition, with the protection of and increase in the number of wild animals, some of these wild animals will inevitably destroy crops and attack livestock. Attention should be paid to minimizing the conflicts between people and wild animals to ensure the safety of residents. At the same time, appropriate economic compensation should be given to villagers by state regulations for accidents caused by crucial protected wild animals to improve human enthusiasm for wildlife protection.

Human disturbance limits the expansion of the suitable area for leopards in Ningxia, and the distance from a road and the distance from a settlement are the most critical factors affecting the suitable habitat of leopards on the geographical scale in Ningxia. In practice, human interference involving tourism, digging for medicinal plants, and digging for wild vegetables has a significant impact on the daytime activities of leopards and other wild animals distributed in Liupan Mountain. This needs to be publicized to reduce human activities in the area of leopard activities. Tourists should be advised to travel using prescribed routes and not venture deeply into the interior of the reserve. The road formed by infrastructure restricts the activity area of leopards, and road sounds can disturb leopards. For the core area of the leopard habitat, road construction should be minimized, and mountains should be afforested. The protection of wild animals will increase their populations, and some wild animals will likely destroy crops and attack livestock. Minimizing the conflicts between people and wild animals will help ensure the safety of residents. Additionally, appropriate economic compensation should be given to villagers by state regulations for accidents caused by crucial protected wild animals to ensure positive human responses to wild animal protection.

## 5. Conclusions

The current main factors affecting suitable leopard habitat area were vegetation cover and human disturbance. The most critical factor affecting future suitable habitat area is rainfall. Under the three climate models, the habitat area of the leopard decreased gradually because of an increase in carbon dioxide concentration. In order to better protect the leopard population, it is suggested that the protected area should reduce human disturbance, expand the protection area, promote the restoration of vegetation, and provide a better living environment for the leopard.

## Figures and Tables

**Figure 1 animals-12-01866-f001:**
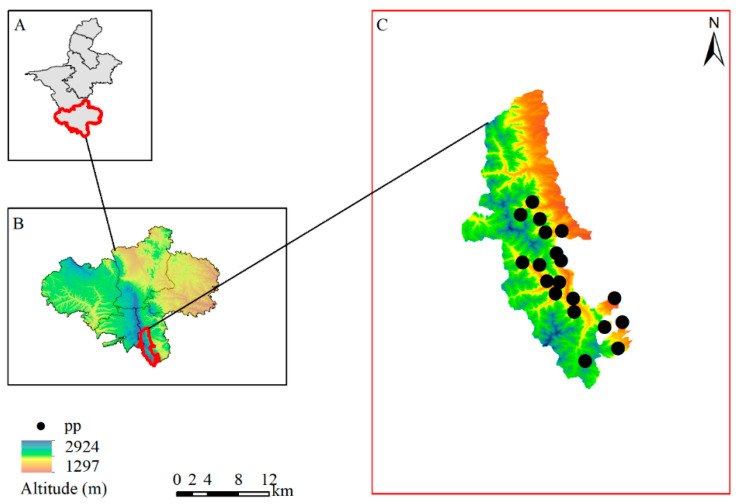
The infrared camera capture distribution site map of leopard. pp: species distribution site of leopards in Liupanshan National Nature Reserve. (**A**): Ningxia Hui Autonomous Region; (**B**): Guyuan City; (**C**): Liupanshan National Nature Reserve.

**Figure 2 animals-12-01866-f002:**
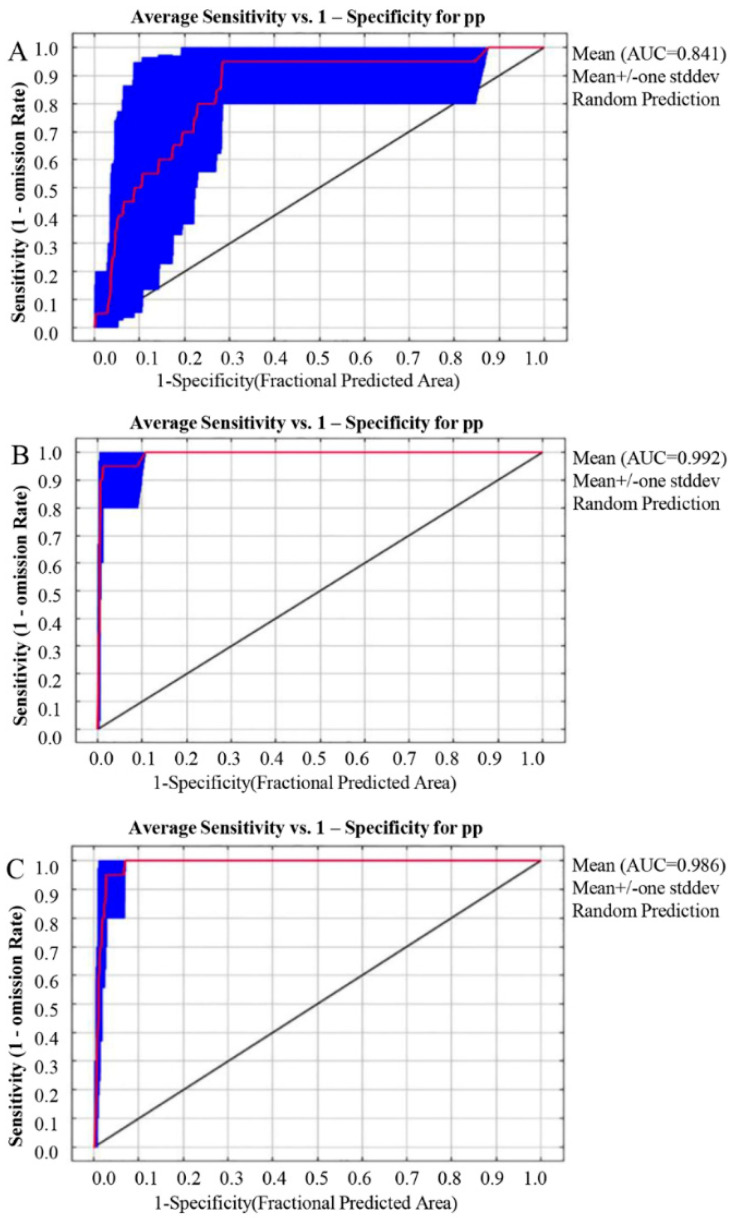
Predicted ROC curves of leopard suitable areas for different geographic scales. (**A**). Predicted ROC curve of the leopard suitable habitat in Liupanshan National Nature Reserve. (**B**). Predicted ROC curve of the suitable leopard area in the Ningxia Hui Autonomous Region. (**C**). Guyuan and the predicted ROC curve of the suitable leopard area.

**Figure 3 animals-12-01866-f003:**
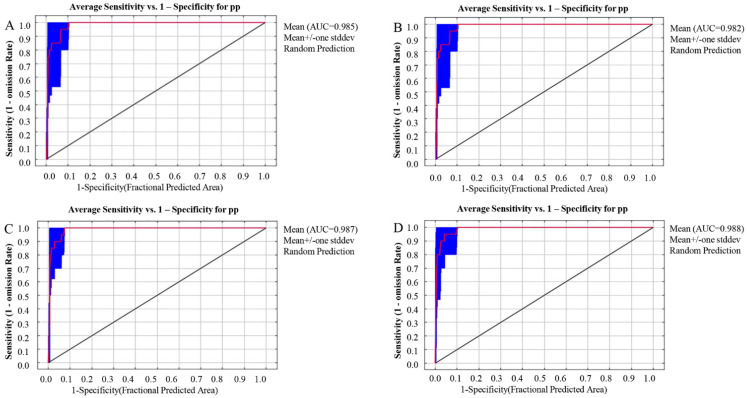
Predicted ROC curve of suitable leopard area under different climate models. (**A**). Predicted ROC curve of the suitable leopard area in modern Guyuan. (**B**). Predicted ROC curve of the suitable leopard area under the future rcp2.6 climate model. (**C**). Predicted ROC curve of the suitable leopard area under the future rcp4.5 climate model. (**D**). Predicted ROC curve of the suitable leopard area under the future rcp8.5 climate model.

**Figure 4 animals-12-01866-f004:**
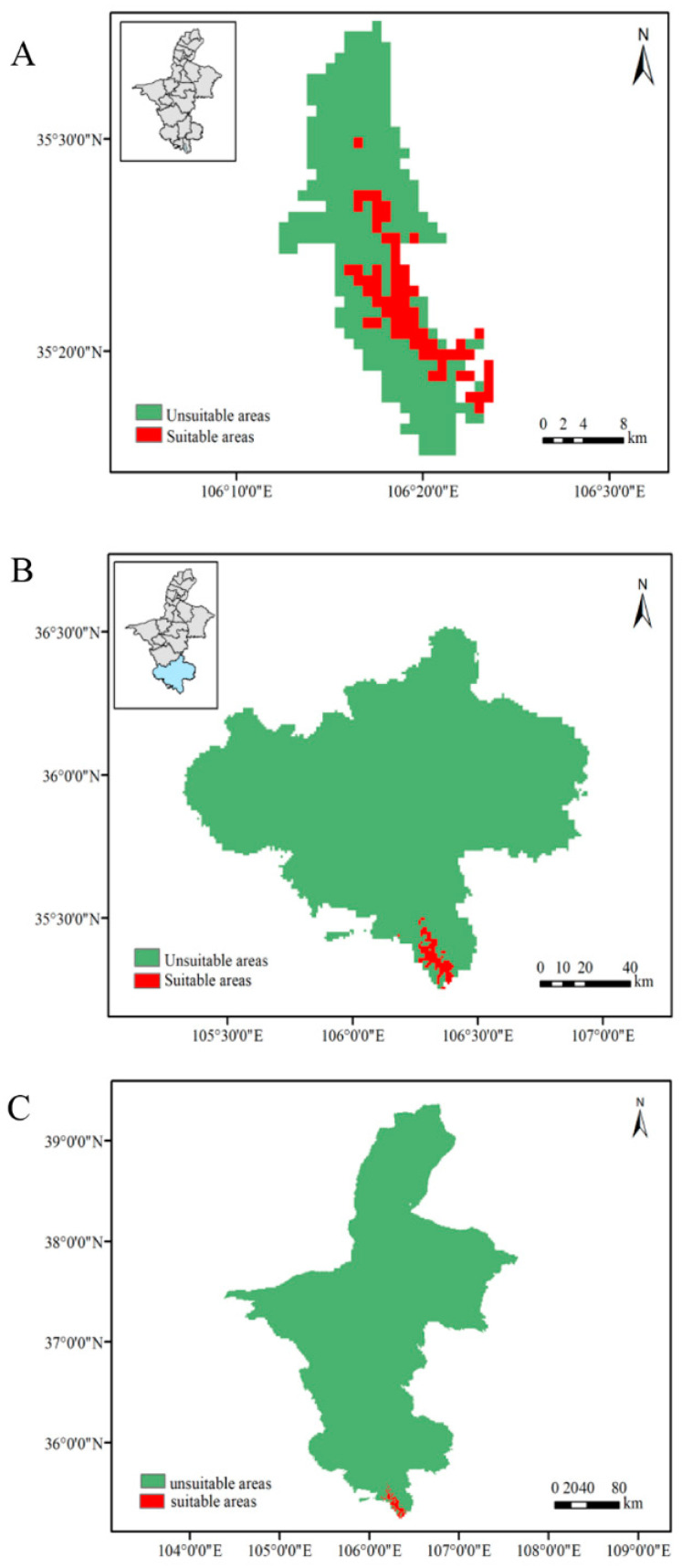
Prediction of the suitable habitat area for leopards at different geographical scales. (**A**). Distribution of suitable habitats for leopards in Liupanshan National Nature Reserve. (**B**). Distribution of suitable habitats for leopards in Guyuan. (**C**). Distribution of suitable habitats for leopards in Ningxia.

**Figure 5 animals-12-01866-f005:**
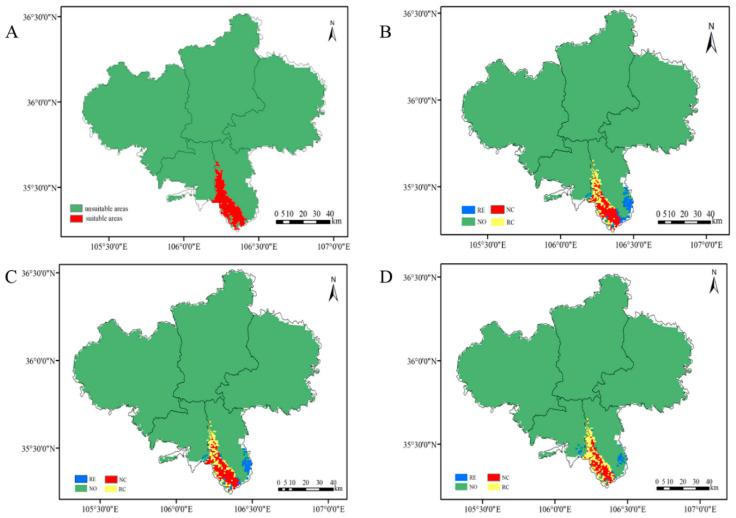
Prediction of changes in suitable habitat areas of leopards under different climate models. (**A**). The modern leopard is suitable for habitat distribution. (**B**). The leopard is suitable for habitat distribution under the future rcp2.6 climate model. (**C**). The leopard is suitable for habitat distribution under the future rcp4.5 climate model. (**D**). The leopard is also suitable for habitat distribution under the future rcp8.5 climate model. RE = range expansion; NO = no occupancy; NC = no change; RC = range contraction. Core area is the area where the suitable habitat of the leopard does not change compared with the suitable habitat in modern Guyuan under different climate models in the future. Expanded habitat is the area where the suitable habitat of the leopard in the future under different climate models expands compared with the habitat in modern Guyuan. Shrunk habitat is the area where the suitable habitat of the leopard under different climate models in the future shrinks compared with the habitat in modern Guyuan.

**Figure 6 animals-12-01866-f006:**
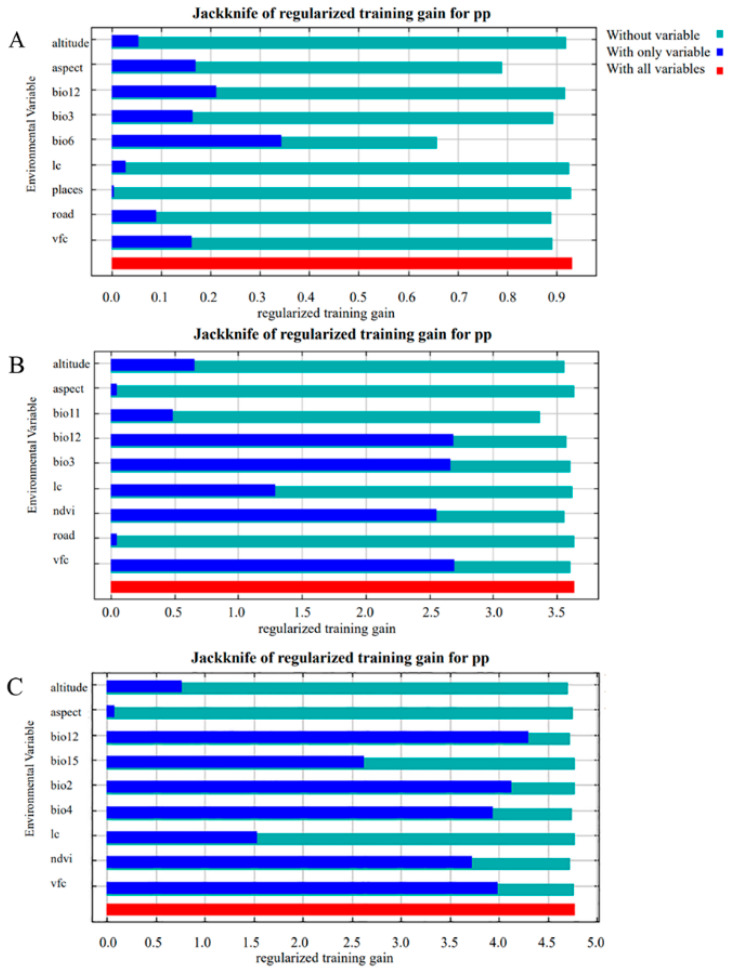
Jackknife method testing the importance of the influence of different geographic scales and main environmental variables on the distribution of suitable leopard habitat. (**A**). The Liupanshan scale environmental variable knife cutting method test. (**B**). The Guyuan city scale environmental variable knife cutting method test. (**C**). The Ningxia city scale environmental variable knife cutting method test.

**Figure 7 animals-12-01866-f007:**
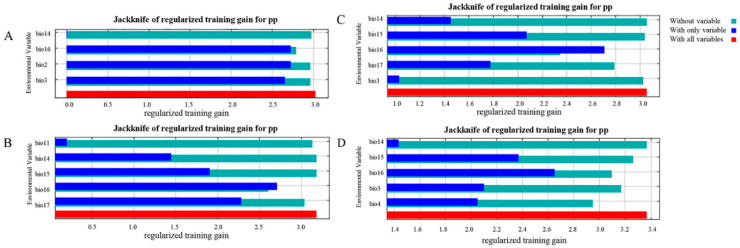
The jackknife method testing the importance of the influence of different climate models’ main environmental variables on suitable leopard distribution. (**A**). Jackknife test of environmental variables in modern situations. (**B**). Jackknife test of environmental variables under the future rcp2.6 climate model. (**C**). Jackknife test of environmental variables under the future rcp4.5 climate model. (**D**). Jackknife test of environmental variables under the future rcp8.5 climate model.

**Table 1 animals-12-01866-t001:** The 28 environmental factors used in the construction of the MAXENT model (including bioclimate, human disturbance, topography, vegetation, and soil factors).

Variable Abbreviation	Variable Description	Unit	Sources
bio 1	annual mean temperature	°C	1
bio 2	mean diurnal range (mean of monthly [max temp − min temp])	°C	1
bio 3	isothermality (bio2/bio7) (×100)	%	1
bio 4	temperature seasonality (standard deviation × 100)	°C/100	1
bio 5	max temperature of warmest month	°C	1
bio 6	min temperature of coldest month	°C	1
bio 7	temperature annual range (bio 5–bio 6)	°C	1
bio 8	mean temperature of wettest quarter	°C	1
bio 9	mean temperature of driest quarter	°C	1
bio 10	mean temperature of warmest quarter	°C	1
bio 11	mean temperature of coldest quarter	°C	1
bio 12	annual precipitation	mm	1
bio 13	precipitation of wettest month	mm	1
bio 14	precipitation of driest month	mm	1
bio 15	precipitation seasonality (coefficient of variation)	%	1
bio 16	precipitation of wettest quarter	mm	1
bio 17	precipitation of driest quarter	mm	1
bio 18	precipitation of warmest quarter	mm	1
bio19	precipitation of coldest quarter	mm	1
lc	land cover type		2
ndvi	normalized differential vegetation index		2
vfc	vegetation fractional cover	%	2
places	distance to villages	m	3
road	distance to roads	m	3
water	distance to water	m	3
slope	slope	°	4
altitude	altitude	m	4
aspect	aspect	°	4

1: Worldclim. https://www.worldclim.org/, accessed on 7 July 2022; 2: National Geomatics Center of China. https://tc211sz.ngcc.cn/ngcc/, accessed on 7 July 2022; 3: National Oceanic and Atmospheric Administration. https://tc211sz.ngcc.cn/ngcc/, accessed on 7 July 2022; 4: Chinese Academy of Sciences. https://english.cas.cn/research/database/, accessed on 7 July 2022.

**Table 2 animals-12-01866-t002:** Correlation analysis of nine environmental factors on a geographic scale in Liupanshan National Nature Reserve.

	bio3	bio6	bio12	lc	Places	Road	vfc	Altitude
bio6	0.308							
bio12	−0.391	−0.410						
lc	−0.321	−0.364	0.161					
places	0.650	−0.267	−0.409	−0.110				
road	−0.303	−0.343	−0.223	0.448	0.190			
vfc	−0.273	0.005	−0.366	0.435	0.059	0.580		
altitude	−0.269	−0.430	0.442	−0.126	−0.049	−0.211	−0.025	
aspect	−0.215	0.200	−0.175	−0.110	−0.102	0.122	0.108	−0.165

**Table 3 animals-12-01866-t003:** The contribution rate of environmental factors in the MAXENT model under different geographical scales and future climate models.

Environmental Factor	Contribution Rate						
	Natural Reserve	Guyuan	Ningxia	Guyuan (Modern)	rcp2.6	rcp4.5	rcp8.5
bio1							
bio2			3.5	60.3			
bio3	8.3	7.8		25.6	2.5		11.5
bio4			0.8				12.4
bio5							
bio6	45.5						
bio7							
bio8							
bio9							
bio10							
bio11		8.2				1.2	
bio12	2.9	8.4	9.3				
bio13							
bio14				1.4	21.8	18.5	19.9
bio15			5.4		3.8	1.1	8.2
bio16				12.7	62.4	60.4	48
bio17					9.5	18.8	
bio18							
bio19							
lc	1	0.5	0.3				
ndvi		42.3	43				
vfc	15.6	27.6	34.2				
places	0.4						
road	5.3	0.3					
water							
slope							
altitude	2.4	4.8	2.7				
aspect	18.6	0.1	0.8				

ndvi, vfc, and bio12 reached 86.50%, the largest contribution to the main factor affecting the leopard habitat.

## Data Availability

The data used in the analysis can be obtained from the authors on request.

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
