# Peer review of "Effects of Climate Change on the Habitat of the Leopard (Panthera pardus) in the Liupanshan National Nature Reserve of China"

_animals, 2022, doi:10.3390/ani12141866_

Round 1

Reviewer 1 Report

Many thanks to all the authors for the opportunity to review their paper. I enjoyed it.

The authors used MAXENT to provide predictive information useful for modeling leopard (Panthera pardus japonensis) habitat given different climate change regimes attributed to specific variables. These data can be used in future to choose locations for productive direct monitoring and to inform land use plans including establishment of habitat corridors and extension of reserve boundaries. The article is clearly and well written, the figures ably illustrate the variables included in the model, and the conclusions are logically linked to the analysis. I recommend publication with minor revision. My few specific comments follow:

Lines 53-58: I am not an expert on the leopard, and for those like me, it might be helpful to understand that there are 31 provinces, with leopards documented in 19 of them (from Laguardia et al. 2015—your citation number 19). It also took me a bit of time to realize that Guyuan is in Ningxia, a province where leopard had previously been recorded. It might be helpful to indicate the numbers of leopards previously documented in Ningxia in those prior studies (although the reader can also look them up), by way of underscoring the importance of this area for leopard conservation.

Introduction in general: I felt that part of your overall findings is the possibility that habitat that currently is not suitable for leopard could become suitable under some climate change scenarios. If I am correct, you might want to frame that view in the introduction with some literature, assuming any exists for leopard. I found this idea interesting, though, as this change could potentially bring people and cats into more contact/conflict by perhaps more often bringing cats into areas where humans are active.

Lines 97-104: Here where you are discussing how the 25 distribution points were reduced to 19—could you please clarify the practical meaning of those decisions? That is, I think you set the parameters as you did to reduce the likelihood of double counting one leopard as two or more (e.g., the same individual recorded at different locations). Could you please provide a bit more information about why these decisions are being made? And briefly how might those decisions influence the models (for example, if you used 25 points instead of the more conservative 19).

Caption for Figure 5 please consider yellow highlight: Fig. 5. Prediction of changes in suitable habitat areas of leopards under different climate models. 239 A. The modern leopard is suitable for habitat distribution. B. The leopard is suitable for habitat 240 distribution under the future rcp2.6 climate model. C. The leopard is suitable for habitat distribution 241 under the future rcp4.5 climate model. D. The leopard is also suitable for habitat distribution under 242 the future rcp8.5 climate model. Core area is the area where the suitable habitat of the leopard 243 does not change compared with the suitable habitat in modern Guyuan under different climate 244 models in the future. Expanded habitat is the area where the suitable habitat of the leopard in the 245 future under different climate models expands compared with the habitat in modern Guyuan. 246 Shrunk habitat is the area where the suitable habitat of the leopard under different climate models 247 in the future shrinks compared with the habitat in modern Guyuan. Include meaning of NE/RC (blue and yellow colors) in the figure caption.

Lines 400-406 repeat lines 383-389.

Author Response

To Reviewer 1:

Many thanks to all the authors for the opportunity to review their paper. I enjoyed it.

The authors used MAXENT to provide predictive information useful for modeling leopard (Panthera pardus japonensis) habitat given different climate change regimes attributed to specific variables. These data can be used in future to choose locations for productive direct monitoring and to inform land use plans including establishment of habitat corridors and extension of reserve boundaries. The article is clearly and well written, the figures ably illustrate the variables included in the model, and the conclusions are logically linked to the analysis. I recommend publication with minor revision. My few specific comments follow:

Lines 53-58: I am not an expert on the leopard, and for those like me, it might be helpful to understand that there are 31 provinces, with leopards documented in 19 of them (from Laguardia et al. 2015—your citation number 19). It also took me a bit of time to realize that Guyuan is in Ningxia, a province where leopard had previously been recorded. It might be helpful to indicate the numbers of leopards previously documented in Ningxia in those prior studies (although the reader can also look them up), by way of underscoring the importance of this area for leopard conservation.

Answer:Thank you for your advice , the Citation (Shuang, W.G.; Hong, F.Z.; Xiao, M.W.; Xiu, Q.X.; Wei, Z.; Yong, Q.H.; Li, N.S.; Gu, C.X.; Han, H.; Cong, R. G.; Cai, H.Y.; Gui, X.C.; Zhou, Z.; Ying, Z.; Zheng, P.W.; Gao, J.H.; Ming, Z.J.; Shao, X.W.; Yong, A.Z.; Yan, L.Z.; Yue, Q.L.; Hong, J.L.; Rui, W.; Zhuang, S.; Yan, L.; Jun, L.S.; Tao, Z.; Dong, R.Z.; Xue, Q.Z.; Long, C. Z.; Zhi, H.G.; Xu, J.; Qiang, C.; Ju, P.L.; Ying, T. Investigation of wild animal resources such as Leopard in Liupanshan National Nature Reserve of Ningxia. Ningxia Hui Autonomous Region, Ningxia Liupanshan National Nature Reserve Administration. 2019) contains the number of leopards surveyed in the Ningxia Autonomous Region. The number is 29-41

Introduction in general: I felt that part of your overall findings is the possibility that habitat that currently is not suitable for leopard could become suitable under some climate change scenarios. If I am correct, you might want to frame that view in the introduction with some literature, assuming any exists for leopard. I found this idea interesting, though, as this change could potentially bring people and cats into more contact/conflict by perhaps more often bringing cats into areas where humans are active.

Lines 97-104: Here where you are discussing how the 25 distribution points were reduced to 19—could you please clarify the practical meaning of those decisions? That is, I think you set the parameters as you did to reduce the likelihood of double counting one leopard as two or more (e.g., the same individual recorded at different locations). Could you please provide a bit more information about why these decisions are being made? And briefly how might those decisions influence the models (for example, if you used 25 points instead of the more conservative 19).

Answer:Thank you for your comments, to a certain extent, it can be understood in this way. If we use 25 distribution points, some data points will have serious spatial autocorrelation because the distance is too close, which will lead to the model to a certain extent. The possibility of the existence of the golden leopard predicted by the model in this area is improved. Removing similar species distribution points can make the model prediction results more aligned with the actual situation.

Caption for Figure 5 please consider yellow highlight: Fig. 5. Prediction of changes in suitable habitat areas of leopards under different climate models. 239 A. The modern leopard is suitable for habitat distribution. B. The leopard is suitable for habitat 240 distribution under the future rcp2.6 climate model. C. The leopard is suitable for habitat distribution 241 under the future rcp4.5 climate model. D. The leopard is also suitable for habitat distribution under 242 the future rcp8.5 climate model. Core area is the area where the suitable habitat of the leopard 243 does not change compared with the suitable habitat in modern Guyuan under different climate 244 models in the future. Expanded habitat is the area where the suitable habitat of the leopard in the 245 future under different climate models expands compared with the habitat in modern Guyuan. 246 Shrunk habitat is the area where the suitable habitat of the leopard under different climate models 247 in the future shrinks compared with the habitat in modern Guyuan. Include meaning of NE/RC (blue and yellow colors) in the figure caption.

Answer:Thank you for your advice s, we have added the text to the corresponding part of the article and explained the meaning of the figure caption: RE= range expansion; NO=no occupancy; NC=no change; RC=range contraction.

Lines 400-406 repeat lines 383-389.

Answer:Thank you for your comments, we have deleted the repetition.

Reviewer 2 Report

This is a very solid and important paper.  It models current habitat suitability for leopards in three different-scaled regions using camera trap data and environmental data.  It then applies different climate models to project how habitat suitability for leopards will change as a result of climate change.  The conclusions are solid, and the authors' recommendations for leopard conservation are sensible.  

The language of the introduction and discussion was direct and easily understood.  However, the Methods did not adequately identify or explain the different spatial scales examined, or why they were chosen for the analysis.   The maps (starting with Fig. 1) must be labelled to indicate what regions are being described.  The three regional scales are introduced abruptly without any explanation or justification in l. 136-138; it will be essential to describe them at greater length, and explain why they are relevant to leopard conservation.   Perhaps this explanation can be placed in "Study Area." 

Small notes: 

l. 78: Should hm2 be km2? And what is the size of the Liupanshan National Reserve?

l. 100:  30" means 30 inches.  Can that be correct?

l. 147:  The sentence appears to stop in the middle.

I found much of the Results section very difficult to read, even with multiple readings.  It will require revision with someone who is more comfortable with English.  For example, L. 192-195 is nonsensical; the paper uses "ratio" when I think they mean "percent" or "proportion," and from L. 216-248, and also l. 359-364 in the discussion, it is often not clear what number is the numerator and what number is the denominator.  These paragraphs are very important for the reader to be able to understand, because they are the heart of the results, but I could not understand them no matter how carefully I looked at them.

Fig. 5 is also very important, but the color legend is not explained (what are RE, NO, NC, and RC?).  Also the figure labels C and D are omitted from the caption.

More small notes:

l. 286 -- small transcription error with "table."

l. 291 -- I think there's a translation error, with "jackknife" being translated as "knife-cutting method test"

l. 400-415 repeat the previous paragraph and should be deleted.

Author Response

To Reviewer 2:

This is a very solid and important paper.  It models current habitat suitability for leopards in three different-scaled regions using camera trap data and environmental data.  It then applies different climate models to project how habitat suitability for leopards will change as a result of climate change.  The conclusions are solid, and the authors' recommendations for leopard conservation are sensible.  

The language of the introduction and discussion was direct and easily understood.  However, the Methods did not adequately identify or explain the different spatial scales examined, or why they were chosen for the analysis.   The maps (starting with Fig. 1) must be labelled to indicate what regions are being described.  The three regional scales are introduced abruptly without any explanation or justification in l. 136-138; it will be essential to describe them at greater length, and explain why they are relevant to leopard conservation.   Perhaps this explanation can be placed in "Study Area." 

Answer:Thank you for your advice s, we have labeled Ningxia Autonomous region, Guyuan City, and Liupanshan National Nature Reserve on the maps. In addition, Result 3.1, 191-194 explains the reasons for the prediction at three regional scales.

Small notes: 

  1. 78: Should hm2 be km2? And what is the size of the Liupanshan National Reserve?

Answer:Thank you for your comments, hm2 is not km2. The area of Liupanshan National Nature Reserve is 678km2. We have converted hm2 to km2.

  1. 100:  30" means 30 inches.  Can that be correct?

Answer:Thank you for your advice s, 30 "represents a unit of latitude and longitude, not a unit of length. The 30 "’s geographical length should be about 1km, not 30 inches."

  1. 147:  The sentence appears to stop in the middle.

Answer:Thank you for your comments, we have adjusted the sentence.

I found much of the Results section very difficult to read, even with multiple readings.  It will require revision with someone who is more comfortable with English.  For example, L. 192-195 is nonsensical; the paper uses "ratio" when I think they mean "percent" or "proportion," and from L. 216-248, and also l. 359-364 in the discussion, it is often not clear what number is the numerator and what number is the denominator.  These paragraphs are very important for the reader to be able to understand, because they are the heart of the results, but I could not understand them no matter how carefully I looked at them.

Answer:Thank you for your advice, 192-195 is sensical, and this passage explains your question about why we set three different research scales. We have made timely changes to 216-248 percentage/ratio, ratio, and 359-364 sentences. If anything else causes reading inconvenience or other problems, welcome to put it forward.

Fig. 5 is also very important, but the color legend is not explained (what are RE, NO, NC, and RC?).  Also the figure labels C and D are omitted from the caption.

Answer:Thank you for your comments, we have explained the meaning of the color legend: RE= range expansion; NO=no occupancy; NC=no change; RC=range contraction. The title complements the figure labels C and D.

More small notes:

  1. 286 -- small transcription error with "table."

Answer:Thank you for your comments, the question mentioned has been revised.

  1. 291 -- I think there's a translation error, with "jackknife" being translated as "knife-cutting method test"

Answer:Thank you for your comments, the question mentioned has been revised.

  1. 400-415 repeat the previous paragraph and should be deleted

Answer:Thank you for your comments, the repetition has been deleted.

Reviewer 3 Report

Dear authors,

In general, what I have read (Introduction and Methodology), has seemed correct to me at a general level, though it requires improvements before publication. Overall, The level of English is correct (although I would recommend a revision by a translator if the manuscript is published). I have found that the manuscript is not correctly cited. Many parts are not accompanied by citations, and most of the references in your manuscript are from scientists from your country of origin, missing many important international papers. I would also like the authors to better elaborate why Maxent is the best distribution model for these cases, since they have not given any citations and yet there are articles that indicate that other types of models may be the same or even, in some cases, better  than Maxent-based models.

I then proceed to post specific comments throughout the manuscript.

Line 24. Add citation

Lines 26-27. Add citations

Lines 29. Could you add some references to those studies you mention in the sentence?

Lines 34-38. The fact that MaxEnt is one of the most widely used species distribution models globally is not synonymous with it being the best or producing the best results compared to other models. Moreover, you have not added any reference to support this claim. GLMs as the Favorability Function produce equal or even better results. See references attached:

Sillero, N. et al. (2021) ‘Want to model a species niche? A step-by-step guideline on correlative ecological niche modelling’, Ecological Modelling, 456(July). doi: 10.1016/j.ecolmodel.2021.109671.

Romero, D. et al. (2016) ‘Comparison of approaches to combine species distribution models based on different sets of predictors’, Ecography, 39(6), pp. 561–571. doi: 10.1111/ecog.01477.

Lines 47. Please add the scientific name to the species

Lines 61-67. Some parts of this paragraph are missing citations, please add them.

Line 78. The hectometre (hm) is not a widely unit of length in English. I recommend transforming the cipher into hectares (ha), which is a more common unit.

Lines 78-79. This may differ depending on the format of the journal, but generally you should add decimal separators to the numbers (in most English-speaking countries, it is usually 10,000.00).

Lines 80-84: Citations?

Lines 88-89: Add the common names.

Figure 1:  This figure is not referenced anywhere in the manuscript. At the very least it should be in some of the subsections of the methodology. As for the design of the figure itself, the altitude colour scale does not seem to match between the first image and the second. In addition, I would add in the explanatory text about what is depicted in map 1 (I imagine it is Guyuan, but I am not entirely sure). I would also add a compass rose or similar to indicate North.

Table 1: Add a new field to record the sources of the variables (which you have explained in detail in the text, but which should also be in the table). For example, you can give each different source a number and then in the table header specify what it corresponds to as well as the link from where you extracted the information if available. For example: 1: Worldclim (https://www.worldclim.org/data/bioclim.html); 2: Chinese Academy of Sciences (https://english.cas.cn/research/database/), etc.

Lines 143-143: Can you elaborate and cite why you have used this global climate model and not others?

Lines 161-169: In relation to the ROC curve, I suggest you cite this article:

Lobo, J. M., Jiménez-valverde, A. & Real, R. AUC: A misleading measure of the performance of predictive distribution models. Glob. Ecol. Biogeogr. 17, 145–151 (2008).

Author Response

To Reviewer 3:

Dear authors,

In general, what I have read (Introduction and Methodology), has seemed correct to me at a general level, though it requires improvements before publication. Overall, The level of English is correct (although I would recommend a revision by a translator if the manuscript is published). I have found that the manuscript is not correctly cited. Many parts are not accompanied by citations, and most of the references in your manuscript are from scientists from your country of origin, missing many important international papers. I would also like the authors to better elaborate why Maxent is the best distribution model for these cases, since they have not given any citations and yet there are articles that indicate that other types of models may be the same or even, in some cases, better  than Maxent-based models.

I then proceed to post specific comments throughout the manuscript.

Line 24. Add citation

Answer:Thank you for your suggestion, thanks for the advance. We have added appropriate references.

Sheehy, J.; Taylor, C.M.; McCann, K.S.; Norris, D.R. Optimal conservation planning for migratory animals: integrating demographic information across seasons. Conservation Letters 2010, 3, 192-202, doi:10.1111/j.1755-263X.2010.00100.x.

Lines 26-27. Add citations

Answer:Thank you for your comments, we have added appropriate references.

Roman-Palacios, C.; Wiens, J.J. Recent responses to climate change reveal the drivers of species extinction and survival. Proc Natl Acad Sci U S A 2020, 117, 4211-4217, doi:10.1073/pnas.1913007117.

Lines 29. Could you add some references to those studies you mention in the sentence?

Answer:Thank you for your suggestion, we have added appropriate references.

Guisan, A.; Zimmermann, N.E. Predictive habitat distribution models in ecology. Ecol Model 2000, 135, 147-186, doi:10.1016/s0304-3800(00)00354-9.

Lines 34-38. The fact that MaxEnt is one of the most widely used species distribution models globally is not synonymous with it being the best or producing the best results compared to other models. Moreover, you have not added any reference to support this claim. GLMs as the Favorability Function produce equal or even better results. See references attached:

Sillero, N. et al. (2022) ‘Want to model a species niche? A step-by-step guideline on correlative ecological niche modelling’, Ecological Modelling, 456(July). doi: 10.1016/j.ecolmodel.2022.109671.

Romero, D. et al. (2016) ‘Comparison of approaches to combine species distribution models based on different sets of predictors’, Ecography, 39(6), pp. 561–571. doi: 10.1111/ecog.01477.

Answer:Thank you for your comments, in our additional citation (Effects of sample size on the performance of species distribution models), it is shown that the prediction accuracy of the maxent model is significantly better than that of other models in the case of a smaller sample size. In addition, the species distribution points we have obtained are less because the species themselves are endangered species, so they are more suitable for the models with a good prediction effect of fewer species distribution, so we chose the maxent model. The GLM model you mentioned has a better effect when there are more sample distribution points. Thank you for asking. We also quote your recommended serial references.

Lines 47. Please add the scientific name to the species

Answer:Thank you for your suggestion, we have added the species name.

Lines 61-67. Some parts of this paragraph are missing citations, please add them.

Answer:Thank you for your suggestion, we have added appropriate references.

Caro, T.M.; O'Doherty, G. On the Use of Surrogate Species in Conservation Biology. Conservation Biology 1999, 13, 805-814, doi:10.1046/j.1523-1739.1999.98338.x.

Line 78. The hectometre (hm) is not a widely unit of length in English. I recommend transforming the cipher into hectares (ha), which is a more common unit.

Answer:Thank you for your comments, we convert it to a more common area in square kilometers.

Lines 78-79. This may differ depending on the format of the journal, but generally you should add decimal separators to the numbers (in most English-speaking countries, it is usually 10,000.00).

Answer:Thank you for your suggestion, we have added a decimal separator to the longer numbers.

Lines 80-84: Citations?

Answer:Thank you for your suggestion, we have added appropriate references.

Xiang, T.W. Scientific investigation of Liupanshan Nature Reserve. Yinchuan: Ningxia People’s Press 1988, 1-356.

Lines 88-89: Add the common names.

Answer:Thank you for your comments, we've added the common English names of these species.

Figure 1:  This figure is not referenced anywhere in the manuscript. At the very least it should be in some of the subsections of the methodology. As for the design of the figure itself, the altitude colour scale does not seem to match between the first image and the second. In addition, I would add in the explanatory text about what is depicted in map 1 (I imagine it is Guyuan, but I am not entirely sure). I would also add a compass rose or similar to indicate North.

Answer:Thank you for your suggestion, we quoted the figure in line 102 of the original manuscript. We adopted and revised the suggestion for the figure. We added the north compass. The color matching of the two elevation figures is the same. The difference between the sea waves' height and the figure's geographical scale makes the two figures look different in color. We have added the corresponding place names to the figure. We have added the north compass. There is a problem with the output of the figure, and we have added it in time. Thanks for the suggestion.

Table 1: Add a new field to record the sources of the variables (which you have explained in detail in the text, but which should also be in the table). For example, you can give each different source a number and then in the table header specify what it corresponds to as well as the link from where you extracted the information if available. For example: 1: Worldclim (https://www.worldclim.org/data/bioclim.html); 2: Chinese Academy of Sciences (https://english.cas.cn/research/database/), etc.

Answer:Thank you for your comments, we have revised it.

Lines 143-143: Can you elaborate and cite why you have used this global climate model and not others?

Answer:Thank you for your suggestion, first, this climate model is widely used and classic, and many researchers worldwide have recognized it. Secondly, the data of this global climate model is open and easy to obtain.

Lobo, J. M., Jiménez-valverde, A. & Real, R. AUC: A misleading measure of the performance of predictive distribution models. Glob. Ecol. Biogeogr. 17, 145–151 (2008).

Answer:Thank you for your suggestion, thank you for your advice. We have quoted this document.